# Association between Availability of Neighborhood Fast Food Outlets and Overweight Among 5–18 Year-Old Children in Peninsular Malaysia: A Cross-Sectional Study

**DOI:** 10.3390/ijerph16040593

**Published:** 2019-02-18

**Authors:** Kee Chee Cheong, Cheong Yoon Ling, Lim Kuang Hock, Sumarni Mohd Ghazali, Teh Chien Huey, Mohd Khairuddin Che Ibrahim, Azli Baharudin, Cheong Siew Man, Cheah Yong Kang, Noor Ani Ahmad, Ahmad Faudzi Yusoff

**Affiliations:** 1Institute for Medical Research, Jalan Pahang, Ministry of Health, Kuala Lumpur 50588, Malaysia; cheongyl@imr.gov.my (C.Y.L.); limkh@imr.gov.my (L.K.H.); sumarni@imr.gov.my (S.M.G.); chienhuey@imr.gov.my (T.C.H.); khairuddin@imr.gov.my (M.K.C.I.); faudzi@imr.gov.my (A.F.Y.); 2Institute for Public Health, Jalan Bangsar, 50598 Kuala Lumpur, Ministry of Health, Malaysia; azlibaharudin@gmail.com (A.B.); smcheong@moh.gov.my (C.S.M.); drnoorani@moh.gov.my (N.A.A.); 3University Utara Malaysia, 06010 UUM Sintok, Kedah Darul Aman, Malaysia; yong@uum.edu.my

**Keywords:** overweight, geospatial analysis, children, fast food

## Abstract

A growing number of fast-food outlets in close proximity to residential areas raises a question as to its impact on childhood overweight and obesity. This study aimed at determining the relationship between the availability of fast-food outlets that were in close proximity to residential areas and overweight among Malaysian children aged 5 to 18 years. Measurement data on the weight and height of 5544 children (2797 boys, 2747 girls) were obtained from the National Health and Morbidity Survey 2011. Overweight (including obesity) is defined as BMI-for-age z-score > +1 SD based on the WHO growth reference. Geographic information system geospatial analysis was performed to determine the number of fast-food outlets within 1000 m radius from the children’s residential address. Multiple logistic regression was conducted to examine the association between the availability of fast-food outlets (none or more than one outlet) and overweight with adjustment for age, sex, ethnicity, monthly household income, parental educational level, residential area and supermarket density. Our results showed that the prevalence of overweight was 25.0% and there was a statistically significant association between the density of fast-food outlets and overweight (odds ratio: 1.23, 95% confidence interval: 1.03, 1.47). Our study suggested that the availability of fast-food outlets with close proximity in residential areas was significantly associated with being overweight among children. Limiting the number of fast-food outlets in residential areas could have a significant effect in reducing the prevalence of overweight among Malaysian children.

## 1. Introduction

Metabolic syndrome is a health condition where there is a clustering of cardiovascular risk factors namely central obesity, dyslipidemia, impaired glucose tolerance, high blood pressure, and insulin resistance. There is growing evidence that obesity increases the risk of metabolic syndrome in children [1]. Data from the National Health Morbidity Surveys (NHMS) showed that the prevalence of obesity among Malaysian children aged less than 18 years had increased dramatically from 6.1% in 2011 [2] to 11.9% in 2015 [3]. Obesity tends to track into adulthood i.e., obese children are more likely to become obese adults [4]. Hence, it is important to tackle obesity early.

Obesity is the result of complex interactions between social, economic, physical environment, genetic and lifestyle factors [5]. Obesogenic environments such as the presence of fast-food outlets close to residential areas may influence children’s dietary pattern toward high consumption of fast food [6]. Fast food has been defined as food that is quickly prepared and served with limited waiting time [7,8]. They are typically high in fat, sodium and sugar, low in fibre and poor in essential micronutrients. Furthermore, they tend to be consumed in large quantities and accompanied by soft drinks which may result in excessive calorie intake and subsequently increases the risk of overweight and obesity [9,10]. The National School-based Nutrition Survey 2012 reported that 82.5% of Malaysian adolescents consumed fast food at least once a week [11]. Another national survey showed that 7% of Malaysian children aged 13–18 years consumed more than two cups of carbonated drinks per day, 6.7% consumed more than three cups of sugar-added drinks per day and 96.3% have inadequate daily intake of fruits and vegetables [2]. These findings have raised concern that excessive fast-food consumption may be contributing to the increasing trend of overweight and obesity among children. One obesity prevention strategy is to control the density of fast-food outlets in the vicinity of residential areas to discourage fast-food consumption [12]. Reviews of previous studies have found inconsistent associations between the availability of fast-food outlets and obesity among children [8,13].

Malaysia is a middle income country with a thriving fast-food industry [14]. Whether the increasing availability of fast foods contributes to obesity in Malaysian children is unknown. Therefore, this study aimed to determine the relationship between the availability of neighborhood fast-food outlets and overweight among Malaysian children controlling for sociodemographic factors. The availability of supermarkets in the neighborhood has been linked with making healthier food choices and lower risk of overweight [15,16]. Therefore, we also hypothesize that the presence of supermarkets in the neighborhood has an inverse relationship with childhood overweight.

## 2. Materials and Methods

We analyzed data from the NHMS 2011. The NHMS 2011 was a nationwide population-based cross-sectional survey that collected health-related information in order to support the Ministry of Health of Malaysia in reviewing its health priorities, developing program strategies and activities as well as planning its future allocation of health resources [2]. Data collection ran from April to July 2011. A sampling frame was provided by the Department of Statistics, Malaysia based on the National Population and Housing Census of 2010. It consisted of the population of Malaysia divided into states, followed by Enumeration Blocks (EBs) and Living Quarters (LQ). There are 16 states (including three federal territories) and approximately 75,000 EBs in Malaysia. Each EB contains between 80 to 120 LQs with an average population of 500 to 600 people per LQ. Two-stage stratified random sampling was applied in this survey. The EBs were first stratified by state and urban-rural area. The number of EBs selected in each state was proportionate to the population size in the states and the urban-rural areas within each state. A total of 794 EBs (484 in urban and 310 in rural areas) were selected. Twelve LQs were randomly selected from each EB and all members of all households within the selected LQs were enrolled. The NHMS 2011 was registered in National Medical Research Register, Ministry of Heath Malaysia and approved by the Medical Research Ethics Committee of the Ministry (NMRR-10-757-6837).

### 2.1. Study Sample

In this study, we extracted data on all children aged 5 to 18 years from the NHMS 2011 dataset. Children who had severe illnesses such as cancer, congenital heart disease, hemophilia, renal problem, physical and mental disability or other debilitating illnesses that may affect their physical growth were excluded from the analysis.

### 2.2. Instrument

All eligible children had their weights and heights measured according to the standard procedure recommended by the World Health Organization [17]. All measurements were performed by trained nurses and research assistants. For weight measurement, Tanita Personal Scale HD 319 was used, while Body Meter SECA 206 was used in measuring height. Both tools were validated and calibrated on a regular basis to ensure the accuracy and precision of measurement. BMI-for-age z-score values were generated for each child using the WHO Anthroplus software [18]. Overweight (including obesity) was defined as BMI-for-age z-score > +1 SD based on the WHO growth reference for children [18]. Sociodemographic variables were obtained using a structured questionnaire.

### 2.3. Spatial Analysis

We defined fast-food outlet as nationally or internationally-known franchised limited service restaurants that sell quickly prepared food with payment made prior to receiving food and expedited food service with limited waiting time. We defined supermarkets as large corporate-owned franchised food stores selling groceries including fresh produce and meat, which are distinguished from grocery stores and smaller non-corporate owned food stores.

There were two stages in preparing the density of fast-food outlets and supermarkets in the residential areas within 1000 m [7]. Firstly, we obtained the coordinates of the houses of children aged 5 to 18 years, fast-food outlets and supermarkets. Secondly, we performed geographic information system (GIS) spatial analysis to extract the density value. The household coordinates of overweight and non-overweight children were extracted from the database of the National Health and Morbidity Survey 2011 [2]. The coordinates of the fast-food outlets and supermarkets were scraped from their respective websites in 2015 using a customized programming script written in jQuery and Ajax techniques based on different javascript structure of each website. We assumed that there were no substantial changes in the distribution and number of these fast-food outlets and supermarkets since 2011 as these fast-food corporations and supermarket chains are long-established in Peninsular Malaysia.

Spatial analyses were then performed using the Open Source QGIS 2.8.1-Wien software [19] (QGIS Development Team, 2014). All households, fast-food outlets and supermarkets coordinates that were originally in WGS84 coordinate reference system were mapped and converted to the Kertau (RSO)/RSO Malaysia (m) coordinate reference system. The 1000 m buffer zones surrounding each house address were first generated with GIS buffer analysis. Then, the number of fast-food outlets and supermarkets within the buffer zones were calculated using the GIS Points in Polygon tool.

### 2.4. Statistical Analysis

All statistical analyses were performed using SPSS version 20.0. Descriptive statistical analysis was conducted to describe the children’s socio-demographic characteristics, including residential area, gender, ethnicity, age group and educational level. The density of fast-food outlets and supermarkets in the residential areas within 1000 m radius from the residential address reported by the respondents were also analysed. Pearson’s chi-squared test and simple logistic regression were conducted to determine the association between overweight and socio-demographic factors and overweight and availability of fast-food outlet and supermarket. Multiple logistic regression was performed to determine the association between fast-food outlet density and overweight with adjustment for residential area, gender, ethnicity, age group, education level, monthly household income and density of supermarkets within 1000 m from residential address. The Hosmer Lemeshow test was applied to assess the goodness-of-fit of the final model. The final model was examined for all potential two-way interactions between the demographic factors.

## 3. Results

A total of 5829 children aged 5 to 18 years (2963 boys and 2866 girls) met the eligibility criteria. Of these, we excluded 151 respondents who had no information on both weight and height measurements, 33 respondents who were missing either weight or height and 7 respondents with no geographic coordinates data. Respondents with BMI-for-age z score >5.0 SD (53 subjects) and <−5.0 SD (31 subjects) were also removed as they were considered outliers [13]. The total sample size in the final statistical analysis was 5544 (2797 boys and 2747 girls). The prevalence of overweight was 25.0%. A total of 945 fast-food outlets and 388 supermarkets were identified in Peninsular Malaysia. The number of fast-food outlets and supermarkets within 1000 m radius of respondents’ residential address ranged between 0–8 and 0–3, respectively. A total of 494 (11.0%) and 1045 children (18.8%) had at least one supermarket and one fast-food outlet within 1000 m radius of their houses, respectively (Table 1).

Figure 1 shows the distribution of fast-food outlets and houses of overweight and non-overweight children. Most fast-food outlets were clustered within densely populated major cities. More than one-fourth (27.6%, 288/1045) of those who had at least one fast-food outlet within 1000 m radius from their homes were overweight.

Pearson’s chi-square tests showed that gender [χ^2^(df) = 11.96(1), *p* = 0.001] age [14.42(2), *p* = 0.001] ethnicity [9.82(3), *p* = 0.002], education level [34.4(3), *p* < 0.001], monthly household income [16.93(3), *p* = 0.001] and number of fast-food outlets within 1000 m radius from the children’s houses [4.5(1), *p* = 0.034] were significantly associated with risk of overweight (Table 1). Multivariable logistic regression analysis showed that children who had at least one fast-food outlet within 1000 m radius of their homes were more likely to be overweight (AdjOR: 1.23; 95% CI: 1.03, 1.47) compared to those that did not after adjustment for the other variables. Our study also revealed that children from middle-income households, males, older and Malay ethnicity were independent risk factors for overweight (Table 2).

## 4. Discussion

Our results showed that the availability of at least one fast-food outlet within 1000 m from the reported residential address was significantly associated with overweight among children aged 5–18 years old. To date there have been no previously published studies from Malaysia, however, there are many from other countries, mainly the United States and European countries [8,13]. A recent cross-sectional study from a large sample of children (approximately 3 million) aged 4–5 and 10–11 in England revealed that children who lived in areas with a high number of fast-food outlets per area unit were more likely to be overweight or obese compared to children with fewer fast-food outlets in their living areas after adjustment for socioeconomic status [6]. Another relatively smaller UK study conducted among 33,594 children aged 3–14 years old in Leeds metropolitan area showed that fast-food density in their neighborhood was closely associated with the prevalence of overweight and obesity among children who stayed in that area. However, no association was observed between the distance from children’s house to the nearest fast-food outlet and the risk of overweight or obesity [20]. In addition, analysis of data from 49,770 children aged 4–18 years who had visited selected pediatric practices (August 2011–August 2012) in eastern Massachusetts had shown that living 0.5 miles (approximately 0.8 km) and less from a fast-food outlet were more likely associated with a higher BMI z-score value compared to living more than 2 miles (~3.2 km) away from a fast-food outlet after adjustment for age, sex, ethnicity and household income [21].

The present findings are further substantiated by a recent prospective study which followed up a national sample of 1577 children from 2006/7 to 2012/13 by researchers from the University of the West of England. Their study suggested that children who lived near to fast-food outlets had about double the risk of weight gain compared to those living further away [22]. Another large prospective study which followed up 944,487 Swedish children for 6 years from 2005 to 2010 concluded that children with access to fast-food outlets in their neighborhoods (measured by presence of at least one fast-food outlet per unit area) had a 14% increase in risk of obesity compared to those living in neighborhoods with no accessibility to fast-food outlets, and the association remains statistically significant after controlling for socio-demographic characteristics, neighborhood-level deprivation (characterized by low education level, low income, unemployment and received social welfares assistance), family history of obesity as well as individual and parental hospitalization [23]. This large prospective study also reported a significant positive association between immediate accessibility to fast food in the neighborhood (presence of at least one fast-food outlet within 1 km buffer zone from home) and risk of obesity. However, the risk association disappeared after controlling for confounders. There are several possible explanations for our findings. Due to increasingly hectic lifestyles, there is a rising trend of eating out of the home and take-away food [24] among Malaysian families, and fast-food outlets near residential areas are the most convenient options. Furthermore, most fast-food outlets are open 24 h and/or provide drive-through and home delivery services. Children are especially fond of fast food, approximately 8 out of 10 Malaysian children aged between 10–18 years old consume fast food at least once a week [11]. We believe that this may be because Malaysian cuisine in general is spicy and not particularly suited to childrens’ delicate taste buds, whereas fast food lacks spiciness.

There are some cultural aspects of fast food consumption that may be peculiar to the Malaysian population. For example, Malaysians generally consume fast food as snacks in addition to the three main meals, resulting in excessive calorie intake and consequently overweight/obesity. Moreover, fast-food outlets also serve as a venue for social gatherings such as family outings and parties/celebrations [25], and some parents would even reward good school performance or behaviour with a fast-food treat. Therefore, proximity to fast-food outlets may contribute to more frequent fast-food consumption. Finally, with outlets located nearby, children are able to frequent it without relying on their parents for transportation.

In contrast, a study conducted by Crawford and colleagues on school children in Melbourne, Australia showed that children with at least one fast food outlet within 2 km radius from their homes had lower BMI. They concluded that an increase in the number of fast-food outlets in the home neighborhood may not be associated with increased fast-food consumption. One possible explanation is that the presence of other types of food retailers that provide relatively healthier food may have diluted the effect of exposure to fast food [26]. Another study which analyzed data from 94,348 New York high school students also showed that the number of fast-food restaurants in students’ neighborhoods was inversely associated with risk of high BMI. Yet, another study using data from the US Food Acquisition and Purchase Survey 2012–2013 involving 3748 children age 2–18 years old, which studied associations between access to different types of retail food stores <1 mile from home and overweight/obesity, showed no significant association between fast-food outlets and risk of overweight/obesity [27]. These studies suggest that the ecology of the entire food environment and not just fast-food outlets in the neighborhood should be examined [28]. Other recently published data also found no association between the presence of fast-food outlets within the residential area and risk of overweight or obesity among 1850 Australian children aged 5–15 years old. On the contrary, children were less likely to be overweight or obese if their residence were near to food outlets selling healthy foods [29]. A systemic review of 46 studies conducted within and outside the US revealed inconsistent associations between the fast-food environment and the prevalence of overweight or obesity among children and adults. These ambiguous results could be probably due to the vast differences in study population, study design, definition of fast food outlet, method used in determining fast food density and proximity, assessment of BMI and country-specific food cultures. Consequently, there is no conclusive evidence to support or refute the presence of a relationship between the fast-food environment and the prevalence of overweight and obesity [13].

Previous studies showed that socio-demographic factors such as age, residential areas, gender, ethnicity and household income were confounders to the association between fast-food outlets in residential areas and overweight among children, which we have controlled for [13]. Our results demonstrated that children who were male, older, Malay and from middle-income households were more likely to be overweight. This could be because males and older children are more likely to frequent fast food outlets than female and younger children as they are outgoing, more autonomous, have greater mobility and eat out more frequently [30]. Fast food is generally viewed as inexpensive in Western countries and fast food outlets are mostly built in low socioeconomic status neighborhoods as demand for fast food is higher in these areas [8]. On the contrary, fast food is deceptively expensive and fast-food outlets are frequently located in more affluent areas in Malaysia as dining at fast-food outlets may not be affordable for children from low-income families as compared to those from middle-income families. Meanwhile, there was no risk association between children from high income households and the prevalence of overweight. This could be because children from high-income households can afford healthier foods which are usually more expensive [31]. A population-based study on the dietary pattern of a sample of 454 Malaysian children aged 12–19 residing in the Malaysian east coast state of Kelantan found that Malay children have greater preference for Western-style foods which are high in processed meats compared to children of other ethnic groups [32]. The National School-based Nutrition Survey 2012 also reported the consumption of fast food was highest among Malay children compared to Chinese and Indian children [11]. This partly explains why Malay children were more likely to be overweight as compared to other ethnic groups.

Previous studies have suggested that there is an inverse relationship between the number of supermarkets or grocery stores in the home neighborhood and the risk of overweight or obesity among both children [6,21] and adults [15,33]. This finding is corroborated by other investigators that showed the presence of supermarkets around households was closely related to lower frequency of dining at fast-food outlets [16], and it was postulated that supermarkets sell healthy food products such as fresh fruits, vegetables and dairy products, which in turn provide a healthier alternative to the residents. This is supported by a study that showed the presence of healthy food outlets within 800 m of households significantly decreases the risk of overweight and obesity among children aged 5–15 years old in Perth, Western Australia after controlling for the number of fast-food outlets in the same vicinity and socioeconomic factors [29]. Similar findings were also reported from a study among public school students in Brazil [34]. However, Gorski Findling et al., found no significant association between access to supermarket and grocery stores and overweight/obesity in a study among children in the US [27]. In our study, the number of supermarkets located within 1-km radius from the residential areas was not associated with lower risk of overweight among Malaysian children. There are several plausible factors that may have diluted this association. Firstly, most fast food outlets in Malaysia are located within malls that also house supermarkets. Secondly, while offering healthy food, supermarkets also sell sugary drinks, salty chips and other junk foods, and thirdly, there are also other local wet markets and grocery stores or mini-markets that sell healthy fresh produce.

This study has a few limitations. The cross-sectional nature of the present study implies only the association between fast-food density in a neighborhood and the risk of overweight among children; it does not imply a causal relationship. Also, the present study did not assess the frequency of visit to fast-food outlets, quantity of fast-food consumption and calorie intake [9]. Therefore, the association between fast-food consumption and the availability of fast-food outlets within the residential vicinity could not be determined [35]. We also did not assess other potential confounding factors that might have undermined the association between the density of fast-food outlets and the risk of overweight such as physical activity level, sedentary behavior, fruit and vegetable consumption and types of food consumed in school canteen and at home. Besides fast food outlets, other factors that likely contribute to overweight in children such as the availability and accessibility (or lack) of sports facilities, recreational park, public transportation, availability of fast food during school commute, number of street hawkers and independent fast-food outlets were not captured in the present study due to a lack of such information [36,37]. In spite of these limitations, the present study was based on national data and was controlled for other important potential confounders.

## 5. Conclusions

Our study suggests that children residing in a residential area with at least one fast-food outlet within 1000 m from their house were more likely to be overweight after controlling for gender, age, ethnicity, household income and locality. There was no association between the availability of supermarkets and overweight. Local city councils and related local planning authorities who grant license to fast-foods outlet operators should regulate the growth of such outlets in residential areas as it could have a significant effect on obesity among Malaysian children.

## Figures and Tables

**Figure 1 ijerph-16-00593-f001:**
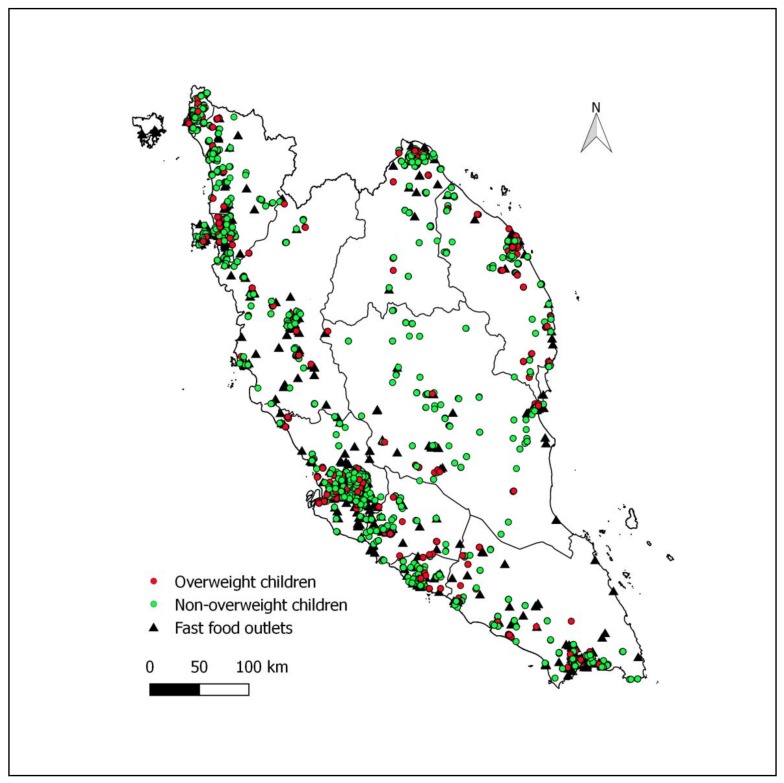
Distribution of fast-food outlets and houses of overweight (BMI-for-age z-score > +1 SD based on the WHO growth reference) and non-overweight children in Peninsular Malaysia.

**Table 1 ijerph-16-00593-t001:** Sociodemographic characteristics of children (*n* = 5544).

	Overweight ^†^			
	No	Yes	Χ^2^ (df) *	*p*-Value
Demographic Characteristics	N (%)	N (%)		
Residential areas				
Urban	2379 (74.2)	827 (25.8)	2.57 (1)	0.109
Rural	1779 (76.1)	559 (23.9)		
Gender				
Male	2042 (73.0)	755 (27.0)	11.96 (1)	0.001
Female	2116 (77.0)	631 (23.0)		
Age group				
5–6	289 (82.1)	63 (17.9)	14.42 (2)	0.001
7–12	1961 (73.3)	714 (26.7)		
12–18	1908 (75.8)	609 (24.2)		
Ethnicity				
Malay	3097 (74.4)	1063 (25.6)	9.82 (3)	0.020
Chinese	637 (77.5)	185 (22.5)		
Indian	349 (73.5)	126 (26.5)		
Others	75 (86.2)	12 (13.8)		
Educational level				
No formal education	21 (77.8)	6 (22.2)	34.4 (3)	<0.001
Primary	2160 (72.0)	842 (28.0)		
secondary	741 (80.2)	183 (19.8)		
Others	1236 (77.7)	355 (22.3)		
Monthly household income (RM)				
<1000	761 (77.5)	221 (22.5)	16.93 (3)	0.001
1000–2999	1689 (76.8)	509 (23.2)		
3000–4999	1704 (72.2)	655 (27.8)		
>5000	4 (80.0)	1 (20.0)		
No. of fast-food outlets within 1000 m radius				
0	3401 (75.6)	1098 (24.4)	4.50 (1)	0.034
≥1	757 (72.4)	288 (27.6)		
No. of supermarket within 1000 m radius				
0	3696 (75.0)	1234 (25.0)	2.77 (3)	0.429
1	355 (75.9)	113 (24.1)		
2	96 (71.6)	38 (28.4)		
3	11 (91.7)	1 (8.3)		

* Pearson’s Chi-squared was performed. df = degree of freedom. ^†^ BMI-for-age z-score > +1 SD (WHO, 2007).

**Table 2 ijerph-16-00593-t002:** Multivariable logistic regression analysis of association between overweight children and availability of fast-food outlets in Peninsular Malaysia (n = 5544).

Demographic Characteristics	COR	95% CI	*p* Value	AdjOR *	95% CI	*p* Value
Residential areas						
Urban	reference			reference		
Rural	0.90	0.80, 1.02	0.109	0.94	0.83, 1.08	0.395
No. of fast-food outlets within 1000 m radius						
0	reference			reference		
≥1	1.18	1.01, 1.37	0.034	**1.23**	**1.03, 1.47**	**0.022**
No. of supermarket within 1000 m radius						
0	reference			reference		
1	0.95	0.76, 1.19	0.672	0.85	0.66, 1.08	0.176
2	1.19	0.81, 1.74	0.381	0.94	0.63, 1.41	0.772
3	0.27	0.04, 2.11	0.213	0.24	0.03, 1.86	0.171
Gender						
Female	reference			reference		
Male	1.24	1.10, 1.40	0.001	**1.24**	**1.10, 1.41**	**0.001**
Age group (year)						
5–6	reference			reference		
7–12	1.67	1.26, 2.22	0.001	**1.39**	**1.02, 1.89**	**0.035**
12–18	1.46	1.10, 1.95	0.009	1.38	1.0, 1.90	0.052
Ethnicity						
Malay	reference			reference		
Chinese	0.85	0.71, 1.01	0.066	**0.764**	**0.64, 0.92**	**0.004**
Indians	1.05	0.85, 1.31	0.645	1.02	0.82, 1.27	0.878
Others	0.47	0.25, 0.86	0.015	**0.49**	**0.26, 0.91**	**0.025**
Educational levels						
No formal education	reference			reference		
Primary	1.36	0.55, 3.39	0.504	1.01	0.40, 2.56	0.986
secondary	0.86	0.34, 2.17	0.757	0.64	0.25, 1.64	0.347
Others	1.01	0.40, 2.51	0.991	0.79	0.31, 2.00	0.611
Monthly Household Income (RM)						
<RM1000	reference			reference		
RM1000–2999	1.04	0.87, 1.25	0.686	1.02	0.85, 1.22	0.855
RM3000–4999	1.32	1.11, 1.58	0.002	**1.30**	**1.09, 1.56**	**0.004**
≥RM5000	0.86	0.10, 7.74	0.894	0.93	0.10, 8.51	0.947

AdjOR = Adjusted Odds Ratio. * Adjusted for all other variables in the final model. COR = Crude Odds Ratio, Hosmer Lemeshow test indicated the final model was fit (*p* = 0.207), No two-way interactions were found among the demographic factors in the final model.

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
