# Peer review of "Association between Availability of Neighborhood Fast Food Outlets and Overweight Among 5–18 Year-Old Children in Peninsular Malaysia: A Cross-Sectional Study"

_ijerph, 2019, doi:10.3390/ijerph16040593_

Round 1
Reviewer 1 Report
The introduction of the study is adequate and with updated quotations, however, I suggest that it could be extended a little, in order to support the current problem on which action is going to be taken.
In the same way, I suggest that you reform the objective, because it is too simple in comparison with the results you show.
Likewise, in line 41 (Simmonds et al. 2016), the quotation should not appear, and this quotation has not been found in bibliographical references either. I suggest that you review the text and modify these errors.
The section on material and method must be restructured as it is not clear. I advise you to divide the section into sample, instruments, procedure and data analysis. Otherwise the interpretation for the readers is complex.
In the discussion section, I consider it necessary to include more studies, in order to contrast and support the results found with those that exist to date.
The concussion is too brief, I suggest you broaden it by showing the conclusions you have reached after carrying out this study.
Author Response
Response to Reviewer 1 Comments
Point 1: The introduction of the study is adequate and with updated quotations, however, I suggest that it could be extended a little, in order to support the current problem on which action is going to be taken.
Response 1 : We have extended the last 2 paragraphs of the introduction.
Point 2: In the same way, I suggest that you reform the objective, because it is too simple in comparison with the results you show.
Response 2 : We have reformed the objective as suggested.[Line 64-67]
Point 3: Likewise, in line 41 (Simmonds et al. 2016), the quotation should not appear, and this quotation has not been found in bibliographical references either. I suggest that you review the text and modify these errors.
Response 3 : All references have been checked and corrected.
Point 4: The section on material and method must be restructured as it is not clear. I advise you to divide the section into sample, instruments, procedure and data analysis. Otherwise the interpretation for the readers is complex.
Response 4: We have divided materials and methods into study sample, instruments, spatial analysis and statistical analysis subsections.
Point 5: In the discussion section, I consider it necessary to include more studies, in order to contrast and support the results found with those that exist to date.
Response 5 : We have added 6 more studies in the discussion.
Point 6: The concussion is too brief, I suggest you broaden it by showing the conclusions you have reached after carrying out this study.
Response 6 : We have added more details to the conclusion.

Reviewer 2 Report
Dear Authors,
Thank you for allowing me to review your manuscript, "Association between availability of neighborhood fast food outlets and overweight among 5-18 year-old children in Peninsular Malaysia: A cross-sectional study". I found it clearly written, well expressed, and engaging. I have only a few comments that I would like to see addressed.
To me, the interesting part about your analysis is that it presents a different social/geographic region in which to examine an already exhaustively studied phenomenon. In fact, this study being done in the US or Europe would be uninteresting for all the reasons that you so clearly pointed out in your discussion - why do the same study that fails to address the threats to causality that has already been done and failed to provide useful, actionable evidence? Thus, I would have appreciated two things: 1. A short discussion of what makes Malaysia unique that may alter the relationshop between fast food and obesity in this context. In your discussion you mention that fast food, though considered cheap eats in the US and Europe, are out of reach for many Malaysians. This, and perhaps a cultural argument that differentiates eating habits in Malaysia from your US and European counterparts would go far in justifying a cross-sectional approach with limited data. 2. A short literature review of food environment research specific to the Malaysian context. It may be that there has been very limited food environment research done in Malaysia (I actually came into this review hoping to get an answer to that question). It may also be the case that there is no reason to suppose that the relationship between fast food and obesity is different in Malaysia than in the US or Europe. However, then I fail to see the value in publishing yet another cross-sectional study that cannot address the weaknesses of previous work. But I don't think that is the case, and I think that comparative nature of this paper, if made more explicit, will make it a valuable contribution to the literature.
Given that you present evidence from multivariable regressions, I'm not sure that X-squared results are very interesting. However, in the regression results, it would be helpful to the reader if you used bolding or starring to identify those Odds Ratios that are significant.
Finally, Figure 1 gives no information that is not also presented on Figure 2; thus, Figure 1 should be eliminated.
Author Response
Response to Reviewer 2 Comments
Point 1: Thank you for allowing me to review your manuscript, "Association between availability of neighborhood fast food outlets and overweight among 5-18 year-old children in Peninsular Malaysia: A cross-sectional study". I found it clearly written, well expressed, and engaging. I have only a few comments that I would like to see addressed.
To me, the interesting part about your analysis is that it presents a different social/geographic region in which to examine an already exhaustively studied phenomenon. In fact, this study being done in the US or Europe would be uninteresting for all the reasons that you so clearly pointed out in your discussion - why do the same study that fails to address the threats to causality that has already been done and failed to provide useful, actionable evidence? Thus, I would have appreciated two things:
1. A short discussion of what makes Malaysia unique that may alter the relationshop between fast food and obesity in this context. In your discussion you mention that fast food, though considered cheap eats in the US and Europe, are out of reach for many Malaysians. This, and perhaps a cultural argument that differentiates eating habits in Malaysia from your US and European counterparts would go far in justifying a cross-sectional approach with limited data.
Response 1: We have added some discussion on the cultural aspects of fast food consumption that may be peculiar to the Malaysian population
Point 2: 2. A short literature review of food environment research specific to the Malaysian context. It may be that there has been very limited food environment research done in Malaysia (I actually came into this review hoping to get an answer to that question). It may also be the case that there is no reason to suppose that the relationship between fast food and obesity is different in Malaysia than in the US or Europe. However, then I fail to see the value in publishing yet another cross-sectional study that cannot address the weaknesses of previous work. But I don't think that is the case, and I think that comparative nature of this paper, if made more explicit, will make it a valuable contribution to the literature.
Response 2: We couldn’t find any fast food environment studies related to child obesity specific to Malaysia, and the few studies in other Asian countries were among adults. We have touched on differences between Malaysia and western countries where we mentioned that fast food is generally viewed as inexpensive in western countries and fast food outlets are mostly built in deprived neighborhoods as demand for fast food is higher in these areas. On the contrary, fast food is deceptively expensive and fast food outlets are frequently located in more affluent areas in Malaysia as dining at fast food outlets may not be affordable for children from low income families as compared to those from middle income families in lines 254-259.
Point 3: Given that you present evidence from multivariable regressions, I'm not sure that X-squared results are very interesting. However, in the regression results, it would be helpful to the reader if you used bolding or starring to identify those Odds Ratios that are significant.
Response 3: The chi-squared results represents the crude association of the sociodemographic distribution of the study sample and overweight status. We have highlighted the significant ORs in bold.
Point 4: Finally, Figure 1 gives no information that is not also presented on Figure 2; thus, Figure 1 should be eliminated.
Response 4: We have removed Figure 1 as suggested.

Round 2
Reviewer 1 Report
Following the changes made, I consider this manuscript to be of interest for publication.